

# Pollen-based reconstructions of Holocene climate trends in the eastern Mediterranean region

Esmeralda Cruz-Silva[1,*], Sandy P. Harrison[1], I. Colin Prentice[2], Elena Marinova[3], Patrick J. Bartlein[4], Hans Renssen[5], Yurui Zhang[6]

1: School of Archaeology, Geography & Environmental Science, Reading University, Whiteknights, Reading, RG6 6AH, UK

2: Georgina Mace Centre for the Living Planet, Department of Life Sciences, Imperial College London, Silwood Park Campus, Buckhurst Road, Ascot SL5 7PY, UK

3: Laboratory for Archaeobotany, Baden-Württemberg State Office for Cultural Heritage Management, Fischersteig 9, 78343 Hemmenhofen-Gaienhofen, Germany

4: Department of Geography, University of Oregon, Eugene, Oregon 97403-1251 USA

5: Department of Natural Sciences and Environmental Health, University of South-Eastern Norway, Bø, Norway

6: State Key Laboratory of Marine Environmental Science, College of Ocean & Earth Sciences, Xiamen University, Xiamen, China

*: Corresponding author

## 1  Abstract

There has been considerable debate about the degree to which climate has driven societal changes in the
eastern Mediterranean region, partly through reliance on a limited number of qualitative records of climate
changes and partly reflecting the need to disentangle the joint impact of changes in different aspects of
climate. Here, we use tolerance-weighted Weighted Averaging Partial Least Squares to derive reconstructions
of mean temperature of the coldest month (MTCO), mean temperature of the warmest month (MTWA),
growing degree days above a threshold of 0°C (GDD0) and plant-available moisture, represented by the ratio
of modelled actual to equilibrium evapotranspiration (α) and corrected for past $CO_2$ changes for 71 individual
pollen records from the Eastern Mediterranean region covering part or all of the interval from 12.3 ka to the
present. We use these reconstructions to create regional composites that illustrate the long-term trends in
each variable. We compare these composites with transient climate model simulations to explore potential
causes of the observed trends. We show that the glacial-Holocene transition and the early part of the Holocene
was characterised by conditions colder and drier than present. Rapid increases in temperature and moisture
occurred between ca 10.3 and 9.3 ka, considerably after the end of the Younger Dryas.  Although the time
series are characterised by centennial-to-millennial oscillations, MTCO showed a gradual increase from 9 ka
to the present, consistent with the expectation that winter temperatures were forced by orbitally-induced
increases in insolation during the Holocene. MTWA also showed an increasing trend from 9 ka and reached a
maximum of ca 1.5°C greater than present at *ca* 5 ka, followed by a gradual decline towards present-day
conditions. A delayed response to summer insolation changes is likely a reflection of the persistence of the
Laurentide and Fennoscandian ice sheets; subsequent summer cooling is consistent with the expected
response to insolation changes. Plant-available moisture increased rapidly between 11 and 9.3 ka and
conditions were slightly wetter than today between 9-8 ka, but thereafter α declined gradually. These trends
likely reflect changes in atmospheric circulation and moisture advection into the region, and were probably
too small to influence summer temperature through land-surface feedbacks. Differences in the simulated



trajectory of α in different models highlight the difficulties in reproducing circulation-driven moisture
advection into the eastern Mediterranean.

## 1. Introduction

The Eastern Mediterranean region is a critical region for examining the long-term interactions between climate
and past societies because of the early adoption of agriculture in the region, which has been widely associated
with the rapid warming at the end of the Younger Dryas (Belfer-Cohen and Goring-Morris, 2011). Societal
collapse and large-scale migrations have been associated with climates less favourable to agriculture during
the 8.2 ka event (Weninger et al., 2006) or to major changes in agricultural practices (Roffet-Salque et al.,
2018). Subsequent periods of less favourable climate, particularly prolonged droughts, have been associated
with the fall of the Akkadian empire ca. 4.2 ka (Cookson et al., 2019), and the end of the Late Bronze Age and
the beginning of the Greek Dark Ages ca 3.2 ka (Kaniewski et al., 2013; Drake, 2012). However, the attribution
of changes in human society to climate changes is not universally accepted.  Flohr et al. (2016), for example,
analysed radiocarbon-dated archaeological sites for evidence of societal changes in response to climate
changes in the early Holocene, particularly the 8.2 ka event, and found no evidence of large-scale site
abandonment or migration although there were indications of local adaptations. However, since Flohr et al.
(2016) did not compare the archaeological records to region-specific climate reconstructions, it is difficult to
assess how far local responses might reflect differences in climate between the sites. Even the societal
response to the early Holocene warming appears to have differed across the region (Roberts et al., 2018).
The need to understand the interactions between climate and past societies in the Eastern Mediterranean is
given further impetus because human modification of the landscape has the potential to affect climate directly
through changes in land-surface properties. The degree to which human modifications of the landscape had a
significant impact on global climate before the pre-industrial period is debated (Ruddiman, 2003; Joos et al.,
2004; Kaplan et al., 2011; Singarayer et al., 2011; Mitchell et al., 2013; Stocker et al., 2017), but these impacts
were likely to be more important in regions with a long history of settlement and agricultural activities
(Harrison et al., 2020).
Much of our current understanding of climate changes in the Eastern Mediterranean region is based on the
qualitative interpretation of individual records (e.g. Roberts et al., 2019). Oxygen-isotope records from
speleothems or lake sediments have been used to infer changes in moisture availability through the Holocene
(e.g. Bar-Matthews et al., 1997; Cheng et al., 2015; Dean et al., 2015; Burstyn et al., 2019) as have pollen-
based reconstructions of changes in vegetation (e.g. Bottema, 1995; Denèfle et al., 2000; Sadori et al., 2011).
Pollen records can also be used to make quantitative reconstructions of seasonal temperatures, and
precipitation or plant-available water (Bartlein et al., 2011; Chevalier et al., 2020). Quantitative
reconstructions of past climates have been made for individual records from the Eastern Mediterranean
region (e.g. Cheddadi and Khater, 2016; Magyari et al., 2019), and syntheses of pollen-based quantitative
climate reconstructions have included sites from this region (Davis et al., 2003; Mauri et al., 2015; Herzschuh
et al., 2022). Davis et al. (2003) provided a composite curve of seasonal temperature changes, but not moisture
changes; both summer and winter temperatures showed very little variation (<1°C) through most of the
Holocene. Mauri et al. (2015) is an updated version of the Davis et al. (2003) reconstructions, with more sites
included but showing similarly muted temperate changes in the Eastern Mediterranean region. Herzschuh et
al. (2022) showed more homogenous changes in both temperature and precipitation across the Eastern
Mediterranean region but it is difficult to compare the two reconstructions directly because they used
different reconstruction techniques. None of the existing reconstructions take account of the impact of





changing $CO_2$ levels on vegetation which could potentially affect the reconstructions of moisture variables
(Prentice et al., 2022). Thus, there is a need for well-founded reconstructions of climate, particularly climate
variables that are relevant for human occupation and agriculture, to be able to address questions about the
interactions between climate and society in the Eastern Mediterranean region.
Here, we provide new quantitative reconstructions of seasonal temperature and plant-available moisture for
71 sites from the Eastern Mediterranean region (20°E – 62°E, 29°N – 49°N), including a correction for the
impact of changing $CO_2$ levels on plant-available moisture reconstructions. We use these reconstructions to
document the regional trends in climate from 12.3 ka to the present. We then explore how far these trends
can be explained by changes in external forcing by comparing the reconstructions with transient climate model
simulations.

## 2. Methods

### 2.1.    Modern pollen and climate data

The modern pollen dataset was obtained from version 1 of the SPECIAL Modern Pollen Data Set (SMPDSv1,
Harrison, 2019), which provides relative abundance data from 6459 terrestrial sites from Europe, the Middle
East and northern Eurasia, assembled from multiple public sources or provided by the original authors. The
SMPDS pollen records have been taxonomically standardized, filtered to remove obligate aquatics,
insectivorous species, introduced species, or taxa that only occur in cultivation, and to group taxa with only
sporadic occurrences into higher taxonomic levels (genus, sub-family or family) and consequently provides
relative abundance data for 247 pollen taxa (Supplementary Table 1). We used the 5840 SMPDS sites from the
area between 20°W to 62°E and 29°N and 75°N to construct the training data set (Supplementary Figure 1);
the sampling outside this box is limited and likely not representative of the diversity of the climate gradients.
At sites with multiple modern samples, we averaged the taxon abundances across all samples, to minimise
over-representation of some localities and hence specific climates, in the training dataset. We used the 195
pollen taxa that occurred at more than 10 sites (Supplementary Table 1) to derive climate-abundance
relationships.
We focus on reconstructing bioclimatic variables that fundamentally control plant distribution, specifically
related to winter temperature limits, accumulated summer warmth and plant-available moisture (Harrison et
al., 2010). The bioclimatic data for each modern site was obtained from Harrison et al. (2019), a dataset that
provides estimates of mean temperature of the coldest month (MTCO), growing degree days above a base
level of 0°C (GDD0), and a moisture index (MI) defined as the ratio of annual precipitation to annual potential
evapotranspiration at each modern pollen site, derived using a geographically-weighted regression of version
2.0 of the Climate Research Unit (CRU) long-term gridded climatology at 10 arc minute resolution (CRU CL
v2.0; New et al., 2002). MTCO and GDD0 were taken directly from the data set. Since Harrison et al. (2019) do
not provide mean temperature of the warmest month (MTWA), we calculated this  based on the relationship
between MTCO and GDD0 given in Wei et al. (2021). We derived an alternative moisture index, α, which is the
ratio between modelled actual and equilibrium evapotranspiration, from MI following Liu et al. (2020). MI and
α both provide good indices of plant-available moisture, but since α has a natural limit in wetter conditions it
is more suitable for discriminating differences in drier climates.



## 2.2. Fossil pollen data

The fossil pollen dataset for eastern Mediterranean region was obtained from the Eastern Mediterranean-Black Sea Caspian Corridor (EMBSeCBIO) database (Harrison et al., 2021), which contains information from 187 records from the region between 29°N and 49°N and 20°E and 62°E. (Note this is a more limited region than used for the modern training data set.) We discarded records (a) from marine environments or very large lakes (>500 km$^2$), (b) with no radiocarbon dating, (c) where the age of the youngest pollen sample was unknown, (d) where there is an hiatus after the youngest radiocarbon date, (e) where more than half of the radiocarbon dates were rejected by the original authors, and (f) where more than half of the ages were based on pollen correlation with other radiocarbon-dated records. However, we kept records where there is an hiatus but where there are sufficient radiocarbon dates above the hiatus to create an age model for the post-hiatus part of the record. We constructed new age models for all the remaining sites (121) using the IntCal20 calibration curve (Reimer et al., 2020) and the 'rbacon' R package (Blaauw et al., 2021) in the framework of the 'AgeR' R package (Villegas-Diaz et al., 2021). Some of these records have no modern samples, where modern was defined as 0-300 yr BP, and thus could not be used to calculate climate anomalies. As a result, 71 pollen records (Figure 1; Supplementary Table 2) were used for the climate reconstructions. These records have a mean length of 6594 and a mean resolution of 228 years. The records were taxonomically standardized for consistency with the training dataset.

## 2.3 Climate reconstructions

We used tolerance-weighted Weighted Averaging Partial Least Squares (*fx*TWA-PLS, Liu et al., 2020) regression to model the relationships between taxon abundances and individual climate variables in the modern training dataset and then applied these relationships to reconstruct past climate using the fossil assemblages. *fx*TWA-PLS reduces the known tendency of regression methods to compress climate reconstructions towards the middle of the sampled range by applying a sampling frequency correction to reduce the influence of uneven sampling of climate space, and by weighting the contribution of individual taxa according to their climate tolerance (Liu et al., 2020). Version 2 of *fx*TWA-PLS (*fx*TWA-PLS2, Liu et al., 2023), applied here, uses P-spline smoothing to derive the frequency correction and also applies the correction both in estimating climate optima and in the regression itself, producing a further improvement in model performance relative to version 1 as published by Liu et al. (2020).

We evaluated the *fx*TWA-PLS models by comparing the reconstructions against observations using pseudo-removed leave-out cross-validation, where one site was randomly selected as a test site and geographically and climatically similar sites (pseudo sites) were removed from the training set to avoid redundancy in the climate information inflating the cross-validation. We selected the last significant component (*p*-value ≤ 0.01) and assessed model performance using the root mean square error of prediction (RMSEP). The degree of compression was assessed using linear regression and local compression was assessed by loess regression (*locfit*). Climate reconstructions were made for every sample in each fossil record using the best models and sample specific errors were estimated via bootstrapping. We applied a correction factor (Prentice et al., 2022) to the reconstructions of α to account for the impact of changes in atmospheric $CO_2$ levels on water-use efficiency, which could have impacted the reconstructions during the earliest part of the records. The correction was implemented using the package codos: 0.0.2 (Prentice et al., 2022) with past $CO_2$ concentration values derived from the EPICA Dome C record (Bereiter et al., 2015).



**2.4. Construction of climate time series**
To obtain climate time series representative of the regional trends in climate, we first screened the
reconstructions to remove individual samples with (a) low effective diversity (< 2) as measured using Hill's N2
diversity measure (Hill, 1973), which could indicate low pollen counts or local contamination, and (b) sample-
specific errors above the 0.95 quantile to remove obvious outliers. This screening resulted in the exclusion of
only a small number of individual samples (see Supplementary Figure 2). We then averaged the reconstructed
values in 300-year bins (slightly larger than the average resolution of the records, 228 years) with 50% overlap
with the first bin centred on 150 yr BP, and excluding any bins with only one sample. The binned values of
individual sites were averaged to produce a regional composite of the anomalies for each climate variable,
where the modern baseline was taken as the first 300-yr bin centred on 150 yr BP. These time series were
smoothed using locally weighted regression (Cleveland & Devlin, 1988) with a window width of 1000 years
(half-window width 500 years) and fixed target points in time to highlight the long-term trends. Confidence
intervals ($5^{th}$ and $95^{th}$ percentiles) for each composite were generated by bootstrap resampling by site over
1000 iterations. We examined the impact of the $CO_2$ correction on reconstructed α (Supplementary Figure 3);
this had no major effect on the reconstructed trends except during the earliest part of the record.
**2.5. Climate model simulations**
We compared the reconstructed climate changes with transient climate model simulations of the response to
external forcing, to determine the extent that the reconstructed climate changes reflect changes in known
forcing. We used transient simulations of the response to orbital and greenhouse gas forcing in the later
Holocene from four models participating in the PAleao-Constraints on Monsoon Evolution and Dynamics
(PACMEDY) project (Carré et al., 2021): the MPI (Max Planck Institute) Earth System Model version 1.2
(Dallmeyer et al., 2020), the AWI (Alfred Wegener Institute) Earth System Model version 2 (Sidorenko et al.,
2019), and two versions of the IPSL (Institut Pierre Simon Laplace) Earth System Model. The IPSL and AWI
simulations were run from 6 ka to 1950 CE, the MPI simulation from 7.95 ka to 1850 CE. We used a longer
transient simulation covering the period from 11.5 ka made with the LOVECLIM model (Goosse et al., 2010)
which, in addition to orbital and greenhouse gas forcing, accounts for the waning of the Laurentide and
Fennoscandian ice sheets (Zhang et al., 2016). Finally, we used two transient simulations from 22 ka to present
made using the Community Climate System Model (CCSM3; Collins et al., 2006). Both were forced by changes
in orbital configuration, atmospheric greenhouse gas concentrations, continental ice sheets and meltwater
fluxes, but differ in the configuration meltwater forcing applied after the Bølling warming (14.7 ka). In the first
simulation (TRACE-21k-I: Liu et al., 2009), there was a sustained meltwater flux of ~0.1 Sv from the Northern
Hemisphere ice sheets to the Arctic and North Atlantic until ca 6 ka, and a continuous inflow of water from
the North Pacific into the Arctic after the opening of the Bering Strait. The second simulation (TRACE-21k-II;
He and Clark, 2022) had no meltwater flux during the Bølling warming or the Holocene but applied a flux of ~
0.17 Sv to the North Atlantic during the Younger Dryas (12.9-11.7 ka). The difference in meltwater forcing
results in a much stronger Atlantic Meridional Overturning Circulation during the Holocene in the TRACE-21k-
II simulation compared to the TRACE-21k-I simulation. Details of the model simulations are given in
Supplementary Table 3. The use of multiple simulations allows the identification of robust signals that are not
model-dependent (see e.g. Carré et al., 2021) and also the separation of the effects of different forcings. The
TraCE-21k-I data were adjusted to reflect the changing length of months during the Holocene, (related to the
eccentricity of Earth's orbit and the precession-determined time of year of perihelion), whereas the other
simulations were not. However, this makes little practical difference for the selection of variables used here
(Supplementary Figure 4).



Outputs from each simulation were extracted for the EMBSeCBIO domain (20˚W − 55˚W, 29˚N − 49˚N). MTCO
and MTWA were extracted directly; GDD0 was obtained by deriving daily temperature values from monthly
data using a mean-preserving autoregressive interpolation function (Rymes & Myers, 2001). Daily values of
cloud cover fraction and precipitation were obtained from monthly data in the same way, and used to estimate
MI through the R package smpds (Villegas-Diaz & Harrison, 2022) before converting this to α following Liu et
al. (2020). For consistency with the reconstructed time series, climate anomalies for 30-yr bins for each land
grid cell within the EMBSeCBIO domain were calculated using the interval after 300 yr BP as the modern
baseline. Since the resolution of the models varies (Supplementary Table 3), and in any case is coarser than
the sampling resolution of the individual pollen records precluding direct comparisons except at a regional
scale, we used all of the grid cells within the EMBSeCBIO domain and did not attempt to select grid cells
coincident with the location of pollen data. A composite was produced by averaging the grid cell time series,
which was then smoothed using locally weighted regression (Cleveland & Devlin, 1988) with a window width
of 1000 years (i.e. a half-window width of 500 years) and fixed target points in time. Confidence intervals (5th
and 95th percentiles) for each composite were generated by bootstrap resampling by grid cell over 1000
iterations.

## 3. Results

### 3.1.    Model performance

The assessment of the model through cross-validation showed that it reproduces the modern climate variables
reasonably well (Table 1, Supplementary Table 4). The best performance is achieved by α ($R^2$ = 0.73, RMSEP =
0.15) and MTCO ($R^2$ = 0.73, RMSEP 3.7˚). The models for GDD0 ($R^2$ = 0.69, RMSEP = 880) and MTWA ($R^2$ = 0.63,
RMSEP = 3.22) were also acceptable. The slopes of the regressions ranged from 0.78 (MTWA) to 0.86 (MTCO),
indicating a small degree of compression in the reconstructions (Table 1).

### 3.2.    Holocene climate evolution in the region

Down-core reconstructions showed broadly coherent signals, although there was variation in both the timing
and magnitude of climate changes across the sites, reflecting differences in latitude and elevation (Figures 2,
3, 4). Nevertheless, the records indicated coherent regional trends over the past 12 ky.
Winter temperature showed a cooling trend between 12.3 and 11ka, with reconstructed MTCO ca 8°C lower
than present at 11 ka (Figure 5). There was a moderate increase in MTCO after 11 ka, followed by a more
pronounced increase of ca 5°C between 10.3 and 9.3 ka. Winter temperatures were only ca 2°C lower than
present at the end of this rapid warming phase. MTCO continued to increase gradually through the Holocene,
although multi-centennial to millennial oscillations were superimposed on the general trend.
The initial trends in summer temperature were broadly similar to those in MTCO, with a cooling between 12.3
and 11ka and reconstructed MTWA ca 2°C lower than present at 11 ka (Figure 5). Summer temperature
increased thereafter, albeit with pronounced millennial oscillations, up to ca 5 ka when MTWA was *ca* 1.5°C
higher than present. There was a gradual decrease in summer temperature after 5 ka. The GDD0
reconstructions showed similar trends to MTWA, reaching maximum values around 5 ka when the growing
season was ca 150 degree days greater than today. The subsequent decline in GDD0 was somewhat flatter,
which presumably reflects the influence of still-increasing winter temperatures on the length of the growing
season.
The trends in α differ from the trends in temperature. The driest conditions occur around 11 ka, when α was
0.2 less than present (Figure 5), when summer and winter temperatures were also at their lowest. There was



a rapid and approximately linear increase in α between 11 and 9.3 ka. Conditions slightly wetter than present
(α greater by 0.05–0.075) occurred between 9 and 8 ka. Thereafter there was a gradual and continuous decline
in α towards the present. The correction for the physiological impact of low $CO_2$ prior to 11 ka (Supplementary
Figure 3) resulted in drier conditions between 12 and 11 ka than obtained without the correction, and these
drier conditions persisted until *ca* 10 ka. The reconstructions with and without the correction are not
statistically different between 10 and 5 ka, but the correction produced marginally wetter reconstructions
after 5 ka. However, the trend of gradual decline in moisture availability towards the present is not affected
by the $CO_2$ correction.

### 3.3.    Comparison with climate simulations

The TRACE-21k-I simulation (Figure 6) shows an initial winter warming between 12-11 ka but MTCO is still ca
3°C lower than present at 11 ka. There is a gradual increase in MTCO from 11ka onwards, although with
centennial-scale variability and a more pronounced oscillation corresponding to the 8.2 ka event. The TRACE-
21k-II simulation is initially slightly colder and displays a two-step warming with a peak at 8.5 ka, when MTCO
is ca 1.5°C lower than present. The later Holocene trend is similar to that shown in TRACE-21k-I. The LOVECLIM
simulation produced generally warmer conditions than either of the TRACE simulations: MTCO is ca 2.5°C
lower than present at 11 ka but the two-step warming is more pronounced and peak warming occurs
somewhat later at ca 7.5 ka when MTCO was only ca 0.25°C lower than present (Figure 7). While all three
models show a rapid warming comparable to the reconstructed warming between 10.3 and 9.3 ka, it is clear
that differences in the ice sheet and meltwater forcings affect both the magnitude and the timing of this trend.
The overall magnitude of the warming after 9 ka in the TRACE-21k-I simulation is consistent with the
reconstructions of MTCO (anomalies of 2.4°C and 2.6°C for model and data respectively). The mid- to late
Holocene trend is similar in the PACMEDY simulations (Figure 8) to both TRACE-21k simulations, both in sign
and in magnitude (ca 1°C between 6 ka and present) and both are consistent with the reconstructions (–0.9 ±
0.7°C). The continuous increase of MTCO is consistent with the change in winter insolation. Given the
similarities between the PACMEDY simulations (which only include orbital and greenhouse gas forcing) and
the LOVECLIM and TRACE simulations, which also include forcing associated with the relict Laurentide and
Fennoscandian ice sheets, it seems likely that orbital forcing was the main driver of winter temperatures in
the EMBSeCBIO region during the later Holocene.
The TRACE-21k-I simulation shows peak summer temperatures between 11-9 ka, when MTWA was ca. 3°C
greater than present (Figure 6). The TRACE-21K-II simulations is initially colder than the TRACE-21k-I
simulation and the peak in summer temperatures occurs at 9 ka, when MTWA was ca 2.5°C greater than
present (Figure 6).  The LOVECLIM simulation is warmer than present from 11.5 ka, but peak warming is only
reached at 7.5 ka when MTWA is ca 2°C (Figure 7). All three simulations show a gradual decrease in summer
temperature through the Holocene after this initial peak. This decreasing trend is also seen in the PACMEDY
simulations from 6 ka (or 8 ka in the case of the MPI simulation) onwards (Figure 8) and the magnitude of the
change over this interval (ca 2°C from 6ka onwards) is similar to that shown by the TRACE and the LOVECLIM
simulations. This similarity suggests that the simulated response is a direct reflection of the change in orbital
forcing. However, the reconstructed changes in summer temperature do not show this gradual decline.
Reconstructed MTWA is ca 4°C colder than the model predictions at 9 ka. The reconstructions show a gradual
increase in MTWA from 9 to 5 ka. Changes in reconstructed temperatures at 5 ka are of a similar magnitude
to simulated temperatures at this time (ca 1°C greater than present) although the late Holocene is marked by
a cooling trend as seen in the simulations. Thus, while the simulated late Holocene trend is consistent with
orbital forcing being the main driver of summer temperatures in the EMBSeCBIO region, the early to mid-



Holocene trend is not. Previous modelling studies have suggested that the timing of peak warmth differs in
different regions of Europe and is associated with the impact of the Fennoscandian ice sheet on regions
climates (Renssen et al., 2009; Blascheck and Renssen, 2013; Zhang et al., 2016).  The differences in the timing
of peak warmth in the EMBSeCBIO region in the TRACE-21k-II and LOVECLIM simulations would be consistent
with this argument but suggest that the timing and magnitude are model-dependent. It is therefore plausible
that the reconstructed trend in MTWA at least during the early Holocene reflects the influence of the relict
Laurentide and Fennoscandian ice sheets in modulating the impact of increased summer insolation until the
mid-Holocene. Given that GDD0 is a reflection of both changes in season length, as influenced by winter
temperatures, and summer warming, the difference between simulated and reconstructed MTWA are also
seen in GDD0 trends during the early part of the Holocene (Figure 6).
The simulations do not show consistent patterns for the trend in α. The TRACE-21k-I simulation (Figure 6)
shows a gradual increase, with minor multi-centennial oscillations from 12 ka to present. (Available model
output variables are not sufficient to calculate α for the TRACE-21k-II or LOVECLIM simulations). One of the
PACMEDY simulations (IPSL-CM5) shows an increase from the mid-Holocene (Figure 8) although the simulated
change is an order of magnitude smaller than over the comparable period in the TRACE-21k-I simulation. The
AWI model shows no trend in α over this period; the remaining two models show increasing aridity from the
mid-Holocene to present (Figure 8). These three models are all broadly consistent with the reconstructions
since the reconstructed decrease in α is small. However, the differences in the sign of the trend between the
different models indicates that changes in moisture are not a straightforward consequence of the forcing, but
must reflect model-dependent changes in moisture supply via changes in atmospheric circulation.
Reconstructions of Holocene climates in Iberia have suggested that land-surface feedbacks associated with
changes in moisture availability have a strong influence on summer temperature (Liu et al., 2023). There does
not seem to be strong evidence for this in the EMBSeCBIO region, given the difference in the trends of α and
MTWA and the muted nature of the trend in α.

## 4.  Discussion

The three temperature-related variables, MTCO, MTWA and $GDD_0$, all show relatively warm conditions around
the late glacial/Holocene transition followed by a cooling that was greatest between *ca* 11 and 10 ka. This
pattern is also shown in regional composites (Figure 9) derived from the reconstructions by Mauri et al. (2015)
and Herzschuh et al. (2022). However, the magnitude of the cooling shown in the Mauri et al. (2015) and
Herzschuh et al. (2022) reconstructions is small compared to our reconstructions. The cool interval starts
somewhat later and persists until 9 ka in the Mauri et al. (2015) reconstructions, but this is partly a reflection
of the fact that these reconstructions were only made at 1 ka intervals and thus the transitions are less well
constrained than in either our reconstructions or those of Herzschuh et al. (2022). This cool interval and the
marked warming seen after 10.3 ka in our reconstructions, does not correspond to the Younger Dryas and the
subsequent warming. Although the Younger Dryas is considered to be a globally synchronous event (Cheng et
al., 2020) and is generally considered coeval with Greenland Stadial I (Larsson et al., 2022), it does not appear
to be strongly registered in the EMBSeCBIO region in any of the quantitative climate reconstructions. This is
consistent with earlier suggestions based on vegetation changes that the Younger Dryas was not a clearly
marked feature over much of this region (Bottema, 1995).
We have shown that winter temperatures increased sharply between 10.3 and 9.3 ka, but then continued to
increase at a more gradual rate through the Holocene. This increasing trend is also seen in the Mauri et al.



(2015) reconstructions of MTCO (Figure 9), although the change from the early Holocene to the present is
smaller (ca 0.5–1°C) in these reconstructions than in our reconstructions and Mauri et al. (2015) do not show
marked cooling around 11 ka. Nevertheless, the consistency between the two reconstructions and between
our reconstruction and the simulated changes in MTCO supports the idea that these trends are a response to
orbital forcing during the Holocene.
Our reconstructions show a gradual increase in summer temperature, as measured by both MTWA and GDD0,
from *ca* 10 to 5 ka when MTWA was ca 1°C warmer than present, followed by a gradual decrease towards the
present. This is not consistent with previous reconstructions. Mauri et al. (2015) show an overall increasing
trend from 9 ka to present. The Herzschuh et al. (2022) shows a completely different pattern, with the
maximum in July temperature at ca. 9 ka and an oscillating but declining trend thereafter (Figure 9). These
differences are too large to be caused by differences in the age models applied. They are also unlikely to reflect
differences in sampling, since the number of sites used is roughly similar across all three reconstructions (71
sites versus 67 sites from Herzschuh et al., 2022 and 409 grid points, based on 57 sites, from Mauri et al.,
2015); most sites are common to all three analyses. The differences must therefore be related to the
reconstruction method. Herzshuch et al. (2022) used the regression-based approach, Weighted Average
Partial Least Squares (WA-PLS), that is the basis for our reconstruction technique, fxTWA-PLSv2.  Mauri et al.
(2015) used the modern analogue technique. However, after taking account of differences caused by the
temporal resolution, there is greater similarity between our reconstructions and those of Mauri et al. (2015)
than between either of these reconstructions and the Herzschuh et al. (2022) reconstructions. Several
methodological issues could be responsible for the differences between the three sets of reconstructions, and
in particular the anomalous moisture trends shown by Herzschuh et al. (2022). Specifically, Herzschuh et al.
(2022) used (1) a unique calibration data set for each fossil site based on modern samples within a 2000 km
radius of that site, rather than relying on a single training data set; (2) a limited set of 70 dominant taxa rather
than the whole pollen assemblage; and (3) included marine records from e.g. the Black Sea, which were
excluded in the other reconstructions because they sample an extremely large area and thus are
unrepresentative of the local climate.
Reconstructed MTWA shows a gradual increase through the early Holocene with maximum values of around
1.5°C greater than present reached at ca 5 ka. Previous studies have shown the timing of maximum warmth
during the Holocene in Europe varied regionally and the delay compared to the maximum of insolation forcing
reflected the impact of the Fennoscandian ice sheet (Renssen et al., 2009; Blascheck and Renssen, 2013; Zhang
et al., 2016). Two of the simulations examined here show a delay in the timing of peak warmth, which occurred
ca 9 ka in the TRACE-21k-II simulation and ca 7.5 ka in the LOVECLIM simulation. Although both sets of
simulations include the relict Laurentide and Fennoscandian ice sheets, neither has realistic ice sheet and
meltwater forcing. In the case of the LOVECLIM simulation, for example, the Fennoscandian ice sheet was
gone by 10 ka whereas in reality it persisted until at least 8.7 ka (Patton et al., 2017). Thus, the impact of the
Fennoscandian ice sheet in delaying orbitally-induced warming could have been greater than shown in this
simulation. Nevertheless, the way in which ice sheets and meltwater forcing are implemented varies between
models; models are also differentially sensitive to the presence of relict ice sheets (Kapsch et al., 2022). It
would be useful to examine the influence of the ice sheets on the climate of the EMBSeCBIO region using
transient simulations at higher resolution or regional climate models. It has been suggested that meltwater
was routed to the Black and Caspian Seas via the Dnieper and Volga Rivers during the early phase of
deglaciation (e.g. Yanchilina et al., 2019; Aksu et al. 2022; Vadsaria et al., 2022) and it would also be useful to
investigate the impact of this on the regional climate.



The availability of water is a crucial factor in the viability of early agriculture (Richerson et al., 2001; Zeder,
2011). We have shown that conditions were markedly drier than today (α anomaly ≈ –0.2) around 11 ka but
that moisture availability increased to levels only very slightly higher than today (α anomaly ≈ 0.05–0.075)
between 9 and 8 ka, before declining to present-day levels. The initial increase in plant-available water, as
indexed by α, could have contributed to promoting the viability of agriculture, as suggested by Richerson et
al. (2001). However, subsequent changes are small even at centennial scale (Figure 5). The reconstructed
trends in α are not captured in the simulations. Although influenced by temperature-driven changes in
evaporation, changes in α in the EMBSeCBIO region are likely to be primarily driven by precipitation changes,
which in turn are driven by changes in atmospheric circulation. There are indeed large simulated changes in
atmospheric circulation through the Holocene in e.g. the LOVECLIM simulations (Supplementary Figure 5) but,
as pointed our earlier, differences in the trend of moisture availability between the models imply that  the
nature of the changes in circulation varies between models and thus does not provide a strong basis for
explaining the observed patterns of change in moisture availability. Furthermore, earlier studies, focusing on
the western Mediterranean (Liu et al., 2023), Europe (Mauri et al., 2014) and central Eurasia (Bartlein et al.,
2017), have shown that models have difficulty in simulating the enhanced moisture transport into the Eurasian
continent shown by palaeoenvironmental data during the mid-Holocene and during the late Holocene. Liu et
al. (2023) have argued that enhanced moisture transport into the Iberian peninsula during the mid-Holocene
led to more vegetation cover and increased evapotranspiration and had a significant impact in reducing
growing season temperatures. However, the differences in the trends of summer temperature and plant-
available moisture through the Holocene suggests that this land-surface feedback was not an important factor
influencing summer temperatures in the EMBSeCBIO region.
We have focused on the composite picture of regional changes across the EMBSeCBIO region, in order to
investigate whether these changes could be explained as a consequence of known changes in forcing. The
data set also provides information on the trends in climate at individual sites. These data could be used to
address the question of whether population density or cultural changes reflect shifts in climate (e.g. Weninger
et al., 2006; Drake, 2012; Kaniewski et al., 2013; Cookson et al., 2019; Weiberg et al., 2019; Palmisano et al.,
2021). In addition, it would also be possible to use these data to explore the impact of climate changes on the
environment, including the natural resources available for people (Harrison et al., in press).

## 5.  Conclusions

We have reconstructed changes in seasonal temperature and in plant-available moisture from 12.3 ka to the
present from 71 sites from the EMBSeCBIO domain to examine changes in the regional climate of the eastern
Mediterranean region. We show that there are regionally coherent trends in these variables. The large
increase in both summer and winter temperatures during the early Holocene considerably post-dates the
warming observed elsewhere at the end of the Younger Dryas, supporting the idea that the impact of the
Younger Dryas in the EMBSeCBIO region was muted. Subsequent changes in winter temperature are
consistent with the expected response to insolation changes. The timing of peak summer warming occurred
later than expected as a consequence of insolation changes and likely, at least in part, reflects the influence
of the relict Laurentide and Fennoscandian ice sheets on the regional climate. Drier-than-present conditions
are reconstructed at the beginning of the Holocene, but there is a rapid increase in plant-available moisture
between 11 and 9 ka, which could have promoted agricultural crops. However, changes in plant-available
water during the middle and late Holocene are small even considering centennial-scale variability.




**Data availability.**

Code for the reconstructions of the climatic variables:
https://github.com/esmeraldacs/EMBSeCBIO_Holocene_climate

**Author Contributions**

ECS, SPH, ICP designed the study; EM, SPH and ECS revised EMBSeCBIO database including the construction of new age models; PJB, HR and YZ provided climate model output; ECS performed the analyses; SPH and ECS wrote the first draft of the paper; all authors contributed to the final version.

**Competing Interests**

The authors declare there are no competing interests.

**Acknowledgements.**

We thank members of the SPECIAL team in Reading and from the Leverhulme Centre for Wildfires, Environment and Society for useful discussions about these analyses.

**Financial support.**

ECS and SPH acknowledge funding support from the ERC-funded project GC2.0 (Global Change 2.0: Unlocking the past for a clearer future, grant number 694481) and from the Leverhulme Centre for Wildfires, Environment and Society through the Leverhulme Trust, grant number RC-2018-023.




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



**Figure and Table Captions**

Figure 1. Distribution of pollen records used in the climate reconstructions. The colour coding shows the length of the record.

Figure 2. Time series of reconstructed anomalies of mean temperature of the coldest month (MTCO) for individual records. Entities are arranged by latitude (N-S). Information about the numbered individual sites can be found in Supplementary Table 1.

Figure 3. Time series of reconstructed anomalies of mean temperature of the warmest month (MTWA) for individual records. Entities are arranged by latitude (N-S). Information about the numbered individual sites can be found in Supplementary Table 1.

Figure 4. Time series of reconstructed anomalies of plant available moisture, expressed as the ratio between potential and actual evapotranspiration ($\alpha$), at individual sites. A correction to account for the direct physiological impacts of $CO_2$ on plant growth has been applied to the reconstructed $\alpha$. Entities are arranged by latitude (N-S). Information about the numbered individual sites can be found in Supplementary Table 1.

Figure 5. Composite changes in reconstructed mean temperature of the coldest month (MTCO), mean temperature of the warmest month (MTWA), growing degree days above a base level of 0°C (GDD0), and plant available moisture expressed as the ratio between potential and actual evapotranspiration ($\alpha$). A correction to account for the direct physiological impacts of $CO_2$ on plant growth has been applied to the reconstructions of $\alpha$. The green line is a loess smoothed curve through the reconstruction with a window half width of 500 years; the green shading shows the uncertainties based on 1000 bootstrap resampling of the records. The bottom panel shows the number of records used to create the composite through time.

Figure 6. Simulated regional changes in mean temperature of the coldest month (MTCO), mean temperature of the warmest month (MTWA), growing degree days above a base level of 0°C (GDD0), and plant available moisture expressed as the ratio between potential and actual evapotranspiration ($\alpha$) in the EMBSeCBIO domain from the TRACE-21K-I (green) and TRACE-21K-II (red) transient simulations. It is not possible to calculate changes in $\alpha$ for the TRACE-21K-II simulation from the available data. Loess smoothed curves were drawn using a window half width of 500 years, and the envelope was obtained through 1000 bootstrap resampling of the sequences. The top panel shows the changes in summer and winter insolation (Wm$^{-2}$) at 40° N.

Figure 7. Simulated regional changes in mean temperature of the coldest month (MTCO), mean temperature of the warmest month (MTWA), and growing degree days above a base level of 0°C (GDD0 ) in the EMBSeCBIO domain from the LOVECLIM transient simulation. It is not possible to calculate changes in $\alpha$ for the LOVECLIM simulation from the available data. Loess smoothed curves were drawn using a window half width of 500 years, and the envelope was obtained through 1000 bootstrap resampling of the sequences.

Figure 8. Simulated regional changes in mean temperature of the coldest month (MTCO), mean temperature of the warmest month (MTWA), and growing degree days above a base level of 0°C (GDD0 ) in the EMBSeCBIO domain from the four PACMEDY simulations. The models are: Max Plank Institute Earth System Model (MPI), Alfred Wagener Institute Earth System Model simulations (AWI), Institute Pierre Simon Laplace Climate Model TR5AS simulation (IPSL-CM5) and Institute Pierre Simon Laplace Climate Model TR6A V simulation (IPSL-CM6). Loess smoothed curves were drawn using a window half width of 500 years and the envelope was obtained through 1000 bootstrap resampling of the sequences.



Figure 9. Comparison of regional composites of reconstructed seasonal temperatures from this study with
those derived from Mauri et al. (2015) and Herzschuh et al. (2022). Mauri et al. (2015 provide mean
temperature of the coldest month (MTCO) and mean temperature of the warmest month (MTWA)
reconstructions, which can be directly compared with our reconstructions. Herzschuh et al. (2022) only
provide reconstructions of July temperature. Our reconstructions are shown in blue, reconstructions based on
the Mauri et al. (2015) data set are shown in green, and reconstructions based on the Herzschuh et al.
reconstruction are shown in orange. The solid line is a loess smoothed curve through the reconstruction with
a window half width of 500 years; the shading shows the uncertainties based on 1000 bootstrap resampling
of the records.
Table 1. Leave-out cross-validation fitness of fxTWA-PLSv2 for mean temperature of the coldest month
(MTCO), mean temperature of the warmest month (MTWA), growing degree days above base level 0°C (GDD0)
and plant-available moisture ($\alpha$) with p-spline smoothed fx estimation, using bins of 0.02, 0.02 and 0.002,
showing results for the selected component for each variable. RMSEP is the root-mean-square error of
prediction. p assesses whether using the current number of components is significantly different from using
one component less. The degree of overall compression is assessed by linear regression of the cross-validated
reconstructions onto the climate variable, where b1 and b1.se are the slope and the standard error of the
slope, respectively. The overall compression is reduced as the slope approaches 1. Full details for all the
components are given in Supplementary Table 4.







Figure 1. Distribution of pollen records used in the climate reconstructions. The colour coding shows the
length of the record.

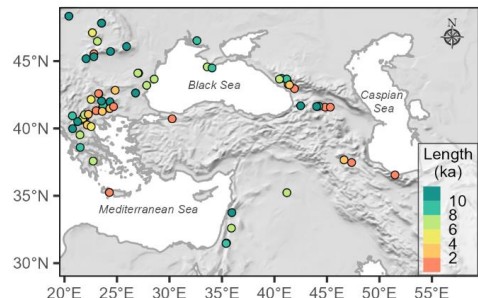











Figure 2. Time series of reconstructed anomalies of mean temperature of the coldest month (MTCO) for individual records. Entities are arranged by latitude (N-S). Information about the numbered individual sites can be found in Supplementary Table 1.

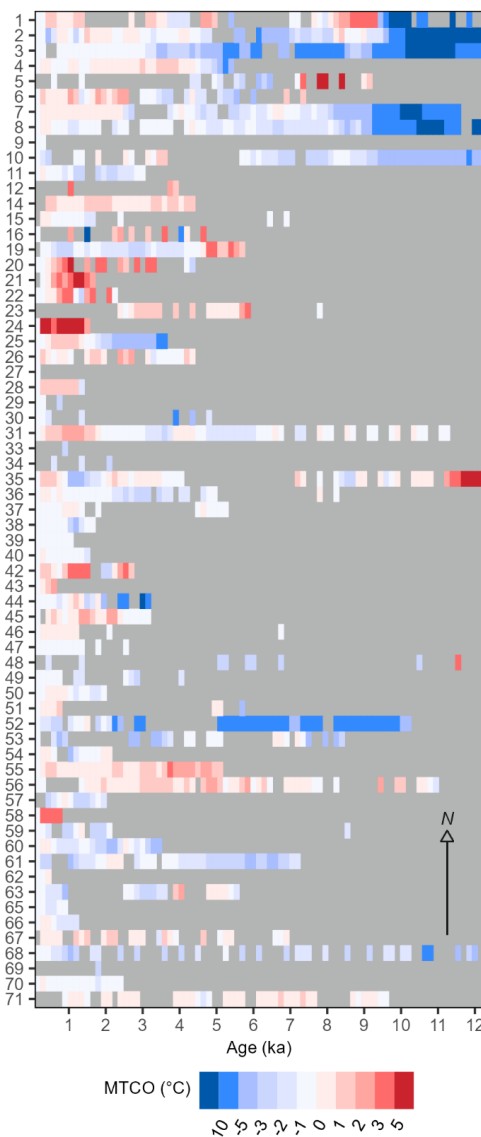



Figure 3. Time series of reconstructed anomalies of mean temperature of the warmest month (MTWA) for individual records. Entities are arranged by latitude (N-S). Information about the numbered individual sites can be found in Supplementary Table 1.

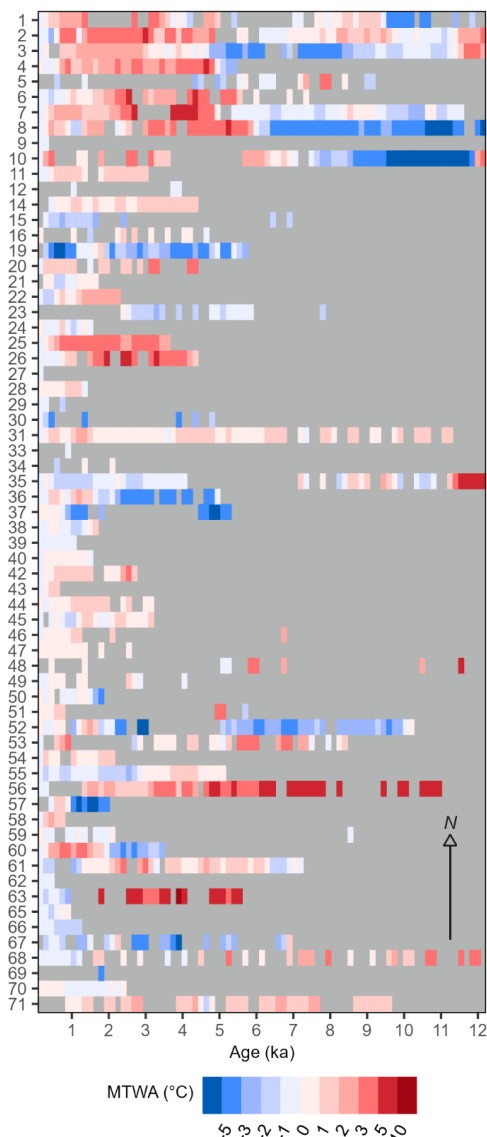



Figure 4. Time series of reconstructed anomalies of plant available moisture, expressed as the ratio between potential and actual evapotranspiration (α), at individual sites. A correction to account for the direct physiological impacts of $CO_2$ on plant growth has been applied to the reconstructed α. Entities are arranged by latitude (N-S). Information about the numbered individual sites can be found in Supplementary Table 1.

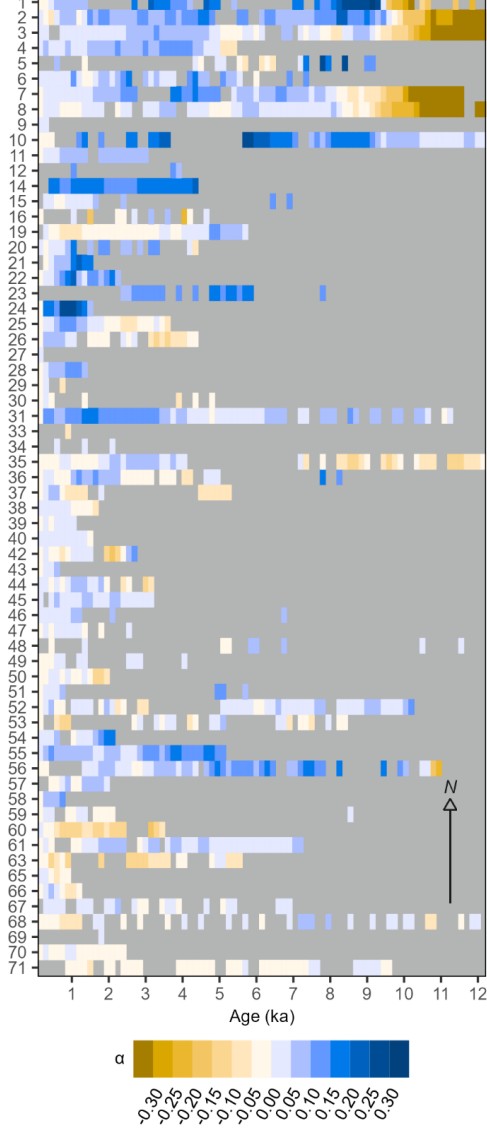



Figure 5. Composite changes in reconstructed mean temperature of the coldest month (MTCO), mean temperature of the warmest month (MTWA), growing degree days above a base level of 0°C (GDD0), and plant available moisture expressed as the ratio between potential and actual evapotranspiration (α). A correction to account for the direct physiological impacts of $CO_2$ on plant growth has been applied to the reconstructions of α. The green line is a loess smoothed curve through the reconstruction with a window half width of 500 years; the blue shading shows the uncertainties based on 1000 bootstrap resampling of the records. The bottom panel shows the number of records used to create the composite through time.

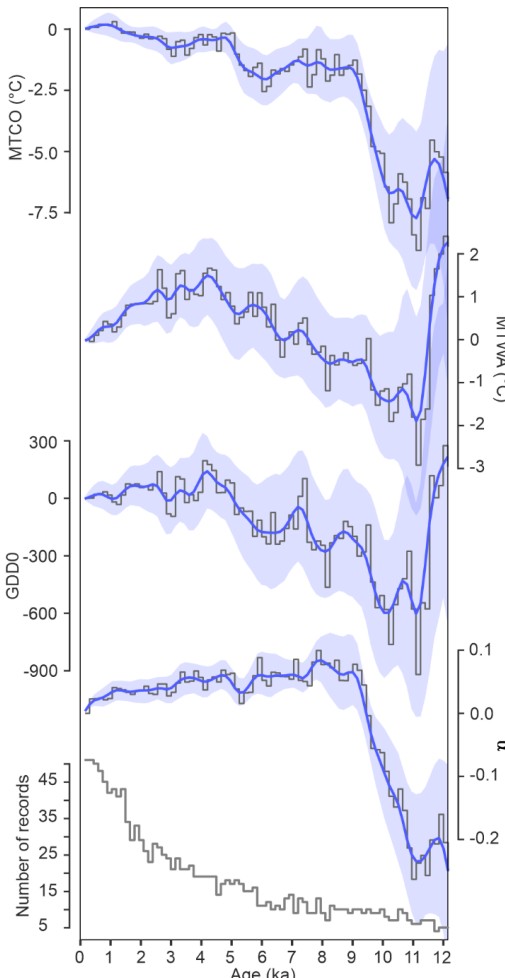





Figure 6. Simulated regional changes in mean temperature of the coldest month (MTCO), mean temperature
of the warmest month (MTWA), growing degree days above a base level of 0°C (GDD0), and plant available
moisture expressed as the ratio between potential and actual evapotranspiration (α) in the EMBSeCBIO
domain from the TRACE-21K-I (green) and TRACE-21K-II (red) transient simulations. It is not possible to
calculate changes in α for the TRACE-21K-II simulation from the available data. Loess smoothed curves were
drawn using a window half width of 500 years, and the envelope was obtained through 1000 bootstrap
resampling of the sequences. The top panel shows the changes in summer and winter insolation (Wm$^{-2}$) at 40°
N.

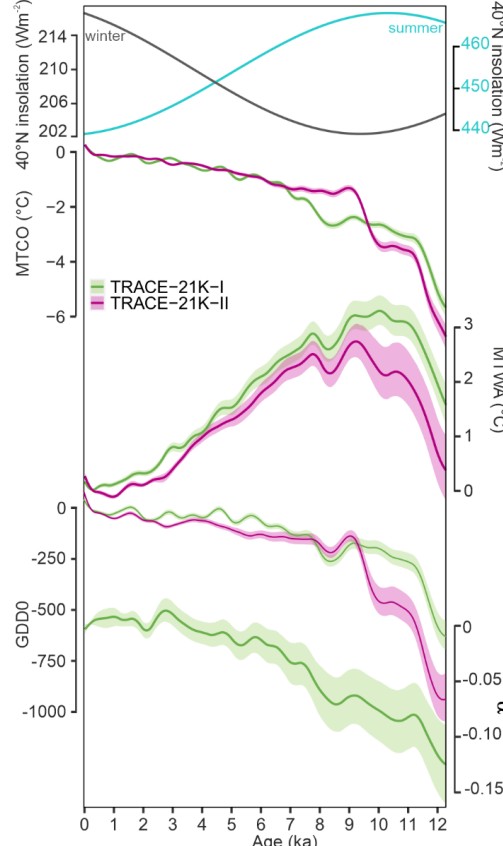













Figure 7. Simulated regional changes in mean temperature of the coldest month (MTCO), mean temperature of the warmest month (MTWA), and growing degree days above a base level of 0°C (GDD0 ) in the EMBSeCBIO domain from the LOVECLIM transient simulation. It is not possible to calculate changes in α for the LOVECLIM simulation from the available data. Loess smoothed curves were drawn using a window half width of 500 years, and the envelope was obtained through 1000 bootstrap resampling of the sequences.

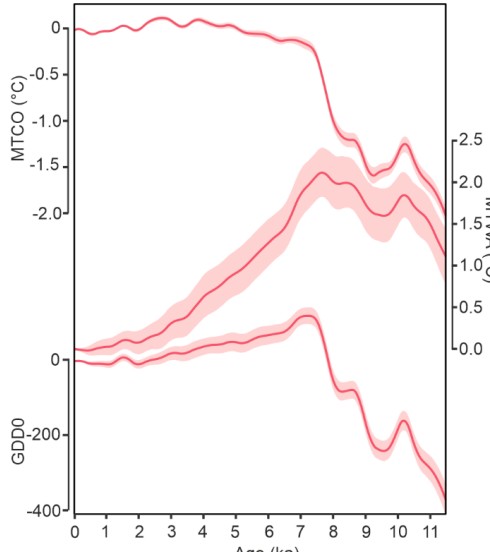



Figure 8. Simulated regional changes in mean temperature of the coldest month (MTCO), mean temperature
of the warmest month (MTWA), and growing degree days above a base level of 0°C (GDD0 ) in the
EMBSeCBIO domain from the four PACMEDY simulations. The models are: Max Plank Institute Earth System
Model (MPI), Alfred Wagener Institute Earth System Model simulations (AWI), Institute Pierre Simon Laplace
Climate Model TR5AS simulation (IPSL-CM5) and Institute Pierre Simon Laplace Climate Model TR6A V
simulation (IPSL-CM6). Loess smoothed curves were drawn using a window half width of 500 years and the
envelope was obtained through 1000 bootstrap resampling of the sequences.

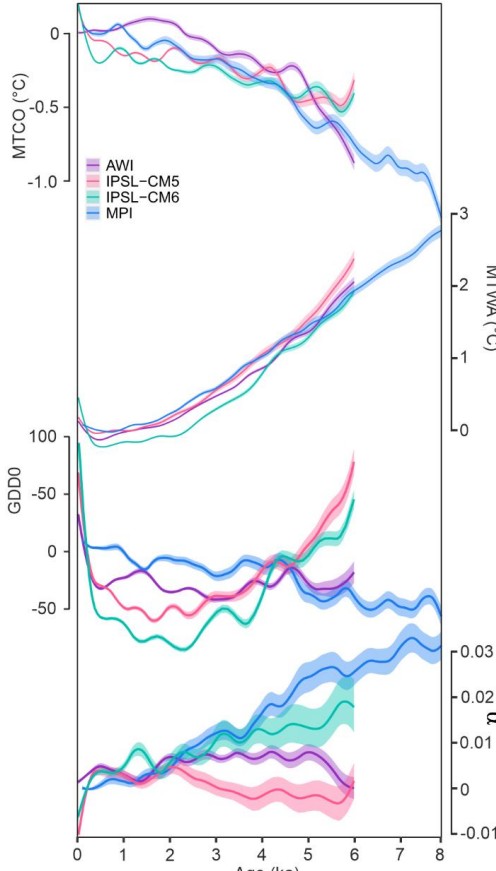











Figure 9. Comparison of regional composites of reconstructed seasonal temperatures from this study with
those derived from Mauri et al. (2015) and Herzschuh et al. (2022). Mauri et al. (2015 provide mean
temperature of the coldest month (MTCO) and mean temperature of the warmest month (MTWA)
reconstructions, which can be directly compared with our reconstructions. Herzschuh et al. (2022) only
provide reconstructions of July temperature. Our reconstructions are shown in blue, reconstructions based on
the Mauri et al. (2015) data set are shown in green, and reconstructions based on the Herzschuh et al.
reconstruction are shown in orange. The solid line is a loess smoothed curve through the reconstruction with
a window half width of 500 years; the shading shows the uncertainties based on 1000 bootstrap resampling
of the records.

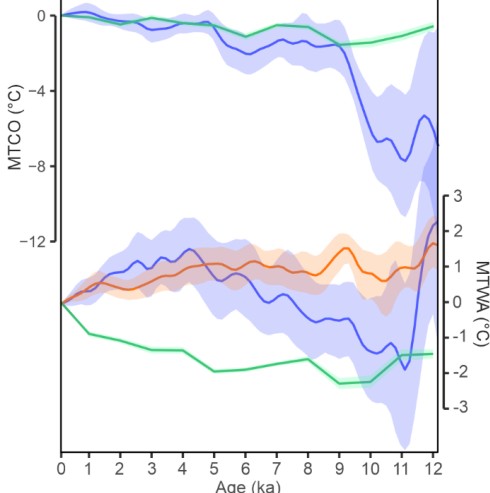






**Table 1.** Leave-out cross-validation fitness of fxTWA-PLSv2 for mean temperature of the coldest month
(MTCO), mean temperature of the warmest month (MTWA), growing degree days above base level 0°C (GDD0)
and plant-available moisture (α) with p-spline smoothed fx estimation, using bins of 0.02, 0.02 and 0.002,
showing results for the selected component for each variable. RMSEP is the root-mean-square error of
prediction. p assesses whether using the current number of components is significantly different from using
one component less. The degree of overall compression is assessed by linear regression of the cross-validated
reconstructions onto the climate variable, where b1 and b1.se are the slope and the standard error of the
slope, respectively. The overall compression is reduced as the slope approaches 1. Full details for all the
components are given in Supplementary Table 4.

| Variable | Selected component | R2 | Average bias | RMSEP | p | b1 | b1.se |
|----------|--------------------|------|--------------|--------|-------|------|-------|
| **MTCO** | 4 | 0.73 | -0.22 | 3.67 | 0.001 | 0.86 | 0.01 |
| **MTWA** | 2 | 0.63 | -0.10 | 3.22 | 0.001 | 0.78 | 0.01 |
| **GDD0** | 2 | 0.69 | 56.46 | 880.33 | 0.001 | 0.79 | 0.01 |
| **α** | 2 | 0.73 | -0.01 | 0.15 | 0.001 | 0.80 | 0.01 |










