# Peer review of "Pollen-based reconstructions of Holocene climate trends in the eastern Mediterranean region"

_EGUsphere, 2023_

## Author Comment (AC1)

**RESPONSE TO RC1 Chris Brierley**

*We thank Chris for his helpful review of our paper, and for providing suggestions about how to make various aspects of the work clearer. Our specific responses are given below (in blue italic) and changes in wording are indicate in normal blue script.*

I found Sect 3.1 a bit of a strange. It is very short and perhaps more importantly provided a substantial jump in meaning and technical depth from the previous section. You never redefine what is meant by "model", so I was stilling thinking of climate models here. You also do not provide much detail about what analysis is being done here. Clearly, this is not the focus of the manuscript. I wonder if it might be more appropriate to subvert this paragraph into the methods section. Alternatively, perhaps you could expand on this section and go explain what analysis is being undertaken at a little more length.

*Our intention here was to provide an evaluation of how well the fxTWA-PLS statistical model performs and hence the reliability of the downcore reconstructions, but we agree that a section entitled model performance coming just after a description of climate model simulations could be potentially confusing and that the text here is terse. We will re-name this section and expand it to make the significance of these results clearer as follows:*

**3.1. Performance of the fxTWA-PLS statistical model**

The assessment of the fxTWA-PLS statistical model through cross-validation showed that it reproduces the modern climate variables reasonably well (Table 1, Supplementary Table 4). The best performance is achieved by $\alpha$ ($R^2 = 0.73$, RMSEP = 0.15) and MTCO ($R^2 = 0.73$, RMSEP 3.7˚). The models for GDD0 ($R^2 = 0.69$, RMSEP = 880) and MTWA ($R^2 = 0.63$, RMSEP = 3.22) were also acceptable. The slopes of the regressions ranged from 0.78 (MTWA) to 0.86 (MTCO), indicating that the degree of compression in the reconstructions in small (Table 1). Thus, the downcore fxTWA-PLS reconstructions of all the climate variables can be considered to be robust and reliable.

The definition of the region of interest seems to vary each time it is written. Please be consistent and accurate. L71 gives a region that seems to extend 10o further east than the final fossil site in Fig 1, and to start further poleward. If there is a change in extent due to the quality controls steps in Sect 2.2, then you ought to discuss this. L189 specifies a region within the Atlantic Ocean, but even after switching oW to oE does not correspond to any previous definition used.

*The original definition (line 71) is the definition of the region as given by the EMBSeCBIO project i.e. 20°E – 62°E, 29°N – 49°N. It is true that we do not have sites beyond 52°E but we do have sites as far north as 49°N in Figure 1. Although we could define the region of interest as less extensive eastward, this might cause confusion with the EMBSeCBIO publications, so we prefer to keep the same definition. We use the same definition in line 109 but reverse the lats and longs, which is unnecessarily confusing. Thanks for pointing out the mistake about E and W in Line 189, which we have now corrected. The area used for the extracting the climate model outputs is somewhat smaller in terms of longitude, to ensure that the grid cells did not include regions outside our pre-defined EMBSeCBIO area of interest and because, as you rightly point out, there are no pollen sites east of 55°E. We should indeed have clarified this. We originally used all the model grid cells within this domain, but have now recalculated the model outputs using land grid cells only (see response to comments on this below). We will modify the text as follows:*

Line 71: Here, we provide new quantitative reconstructions of seasonal temperature and plant-available moisture for 71 sites from the Eastern Mediterranean region (defined by the Eastern Mediterranean-Black Sea-Caspian Corridor, EMBSeCBIO, project as the region between 20°E – 62°E, 29°N – 49°N).

Line 109. ..... from the region between 20°E and 62°E and between 29°N and 49°N.

Line 189 (new line number 191). Outputs from each simulation were extracted for land grid cells in the EMBSeCBIO domain (20°E – 55°E, 29°N – 49°N; this region extends slightly less far eastwards than the EMBSeCBIO region as originally defined but there are no pollen sites beyond 55°E).

The impact of the CO2 correction is noted as having no major effect (L159). I'm not sure I agree after looking at Supplementary figure 3. Although the magnitudes don't alter, there is a 1000 year shift in the timing of the early wetting, which would shift the trend in Fig. 5 to visually appear anti-correlated with MTWA, rather than correlated with MTCO trends.

*Thanks for raising this issue. We re-examined the calculations and realised that we were using mean annual temperature rather than mean growing season temperature as specified in Prentice et al. (2022) in our implementation of the correction. When we implement the α correction using the mean growing season temperature inputs, the reconstructed plant-available moisture is higher as expected and there is no lag in the timing of the wetting compared to the uncorrected curve. We have now corrected Figures 4, 5 and the supplementary Figure 3. We have revised the results in section 3.2 (line 228) as follows:*

The trends in α differ from the trends in temperature. Conditions were similar to present around 11.5 ka (Figure 5). Between 11 and 10 ka, there was a rapid increase in α. Values of α were higher than present (>0.1) between 10 to 6 ka. Subsequently, there was a gradual and continuous decrease in α until the present time. The correction for the physiological impact of $CO_2$ levels was, as expected, largest during intervals when $CO_2$ was lowest (i.e. prior to 11 ka) (Supplementary Figure 4). The reconstructions with and without the correction are not statistically different between 10 and 5 ka, taking account the uncertainties in the reconstructions, but the correction produced marginally wetter reconstructions after 5 ka, with a maximum difference of 0.08. However, the gradually declining trend in moisture availability towards the present is not affected by the $CO_2$ correction.

*Note that we have expanded the discussion of the α trends in response to a comment from Reviewer 2 (see below).*

Fig. 5-9. Please can you consider keeping your temperature records on the same scale. The spacing is changes both within a plot (between MTWA and MTCO) and between the plots.

*We agree that it is not entirely satisfactory to use different scales for the different figures. The difficulty that we have here is that there are marked differences in the range of changes in MTWA and MTCO (Figure 5), and between the simulated temperatures and the reconstructions (Figs 6, 7, 8), and indeed between the three reconstructions (Figure 9). This means that if we use the same ranges for MTWA and MTCO across all of the figures, it will be difficult to see the trends. Since we feel that the similarity of the trends is more important than the magnitude of the changes, particularly between the reconstructions and the simulations, we feel constrained to use different scales to make these more apparent. To improve readability, we have increased the number of scale categories and exclusively utilized whole numbers.*

Fig. 9 needs a key, rather than the colors being provided in the caption.

*We have updated this figure, adding a key indicating the three different sources for the reconstructions.*

Your description of colors in the figure captions can sometimes vary from the colors that I've printed out – e.g. I see no green line on Fig. 5.

*We have checked the descriptions. The caption for Fig. 5 should read "the dark blue line" rather than "the green line" and has been amended.*

Please check the accuracy of your discussion of time trends. I feel that the cooling on L300 should be a warming - as time goes towards the present day.

*We are indeed talking about time progressing towards the present day. The records show relatively warm conditions at the beginning of the reconstructed interval (i.e. the late glacial/Holocene transition) followed by a cooling towards 11ka and conditions remained cooler than before until 10ka after which the reconstructions show warming again. We have tried to make this clearer by amending the text as follows:*

*.... relatively warm conditions around the late glacial/Holocene transition (ca 12 ka) followed by a cooling that was greatest between ca 11 and 10 ka.*

I was confused by the statement about 50% overlap on L152. Do you mean that there is a bin centred on every 150 year interval (so overlapping 50% with the previous bin)? If so, that bin would also have a 100% overlap in total, as it's other half is shared with the next bin. I don't have problem with this approach, but can you please be clearer as to what you are doing.

*Yes indeed we have a 300-year bin centred on every 150 year interval so there is a 50% overlap with the previous interval and a 50% overlap with the subsequent interval. We have tried to make this clearer by rewriting the text as follows:*

*We then averaged the reconstructed values in 300-year bins (slightly larger than the average resolution of the records, 228 years) with 50% overlap. The first bin was centred on 150 yr BP, and subsequent bins were centred at 150 yr increments throughout the record. We excluded any bins with only one sample.*

L339. You make a point about excluding marine records because they are not representative of the local climate. Your regional average does encompass the whole of the Black Sea, You might want to revisit your justification here.

*The statement here is about potential reasons why the reconstruction by Herzschuh et al. differ from those we have made and also the reconstructions made by Mauri et al.   The fact that Herzschuh et al. do not exclude marine records (whereas we do) is one factor that could explain this difference, but it may also be due to other methodological differences as specified. We exclude marine records in making our reconstruction because, according to pollen source area theory, they sample a very large area - in the case of the Black Sea this area is extremely large.*

*We recognise the potential confusion arising from excluding marine records in the reconstructions but not in the model simulation composites. We have now recalculated the model composites using only land grid cells. This does not significantly affect the magnitude or trend shown by the climate simulations, probably due to the models' coarse resolution, and thus this change has no impact on our results or discussion. As described above, we have now replaced the figures and clarified that we are only using land grids in the Methods (see response above).*

L215 describes the earliest portion of the MTCO as having a cooling trend. I feel that a flatline could be drawn through this period in Fig 5.

*It is true that the uncertainties associated with the early part of the record are larger than those after 10 ka, so some caution is required in describing these trends. Neverthless, there is an apparent cooling from 12-11ka and the coldest period is between 11-10ka, after which there is a marked warming. Given that there is a much more pronounced cooling to a minimum around 11ka registered in MTWA and GDD0, we feel that the MTCO trend is probably real. However, we agree that we should be more circumspect here and have rewritten this as follows:*

Winter temperature showed a cooling trend between 12 and 11ka, with reconstructed MTCO ca 8°C lower than present at 11 ka (Figure 5). There was a moderate increase in MTCO after 11 ka, followed by a more pronounced increase of ca 5°C between 10.3 and 9.3 ka. Winter temperatures were only ca 2°C lower than present at the end of this rapid warming phase. There are relatively large uncertainties on the MTCO reconstructions prior to 10.3 ka, so the trends in the early part of the record are not well constrained. However, the phase of rapid warming between 10.3 and 9.3 ka (and the subsequent part of the record) is well constrained. MTCO continued to increase gradually through the Holocene, although multi-centennial to millennial oscillations were superimposed on the general trend.

L299: Please be consistent with the subscript on GDDO

*Thanks for pointing out this inconsistency. We have used GDD0 through most of the manuscript and in the figures, so we have now changed the occasions when this was subscripted for consistency.*

L193: Please spell out Moisture Index

*We defined the moisture index in the Methods section (lines 97-97) as follows "and a moisture index (MI) defined as the ratio of annual precipitation to annual potential evapotranspiration". However, we are happy to spell this out again in the context of the model outputs. We have amended the text as follows:*

and used to estimate the moisture index, MI, i.e. the ratio of annual precipitation to annual potential evapotranspiration, through the R package smpds (Villegas-Diaz & Harrison, 2022) before converting this to α following Liu et al. (2020).

L121: please add years after 6594

*We have amended the text to read:*

have a mean length of 6594 years and a mean resolution of 228 years

L196: Can you please specify temporal and/or spatial resolution?

*The models are all transient, so here we are refering to differences in the spatial resolution. We have amended the text to read:*

Since the spatial resolution of the models varies (Supplementary Table 3),

L230: Please remove "and approximately linear"

*We have removed this phrase.*

L274: regions -> regional

*Thanks for pointing this out. We have made this modification to the text.*

---

## Author Comment (AC3)

**RESPONSE TO RC2 Anonymous Referee #2**

*We thank referee 2 for their comments. Our specific responses are given below (in blue italic) and changes in wording are indicate in normal blue script.*

1. The authors make an intriguing observation about the delay in peak summer warming in their own reconstructions, which they suggest might be attributed to the presence of the Laurentide and Fennoscandian ice sheets. While this hypothesis is plausible, it is crucial for the authors to discuss the limitations and uncertainties of the models used, as the simulations didn't accurately represent the timing and magnitude of ice-sheet and meltwater forcing, especially regarding the Fennoscandian ice sheet.

*We agree that it is important to comment on the limitations and uncertainties associated with the model experiments. We addressed this in our original Discussion (lines 346 to 352) by pointing out that while both the TRACE-2112K-II and the LOVECLIM simulations included the relict Laurentide and Fennoscandian ice sheets, they did not incorporate realistic ice sheet and meltwater forcing. We also pointed to the Kapsch et al. (2022) which shows that differences in ice sheets and meltwater forcing are implemented cause differential sensitivity to the presence of these relict ice sheets. However, we have expanded the text to make this clearer, as follows:*

Reconstructed MTWA shows a gradual increase through the early Holocene with maximum values of around 1.5°C greater than present reached at ca 4.5 ka. Previous modelling studies have shown that the timing of maximum warmth during the Holocene in Europe was delayed compared to the maximum of insolation forcing and varied regionally as a consequence of the impact of the Fennoscandian ice sheet on surface albedo, atmospheric circulation, and heat transport (Renssen et al., 2009; Blascheck and Renssen, 2013; Zhang et al., 2016; Zhang et al., 2023). Two of the simulations examined here show a delay in the timing of peak warmth, which occurred ca 9 ka in the TRACE-21k-II simulation and ca 7.5 ka in the LOVECLIM simulation. Although both sets of simulations include the relict Laurentide and Fennoscandian ice sheets, neither has realistic ice sheet and meltwater forcing. In the LOVECLIM simulation, for example, the Fennoscandian ice sheet was gone by 10 ka whereas in reality it persisted until at least 8.7 ka (Patton et al., 2017). Thus, the impact of the Fennoscandian ice sheet in delaying orbitally induced warming would likely have been greater than shown in this simulation. In addition to differences in the way in which ice sheets and meltwater forcing are implemented in different models, models are also differentially sensitive to the presence of the same prescribed ice sheet (Kapsch et al., 2022). Thus, it would be useful to examine the influence of more realistic prescriptions of the relict ice sheets on the climate of the EMBSeCBIO region using multiple models, and preferably transient simulations at higher resolution or regional climate models. It has been suggested that meltwater was routed to the Black and Caspian Seas via the Dnieper and Volga Rivers during the early phase of deglaciation (e.g. Yanchilina et al., 2019; Aksu et al. 2022; Vadsaria et al., 2022) and it would also be useful to investigate the impact of this on the regional climate.

2. The use of model simulations serves as an important component of this manuscript, providing a means for extrapolating and understanding the potential drivers behind the observed climate trends. The authors might need to provide a more thorough explanation about the discrepancies in the trend of plant-available moisture, α, between the models. They state that the differences likely reflect changes in atmospheric circulation, but don't discuss more what those changes might be, or why

the models differ in their representation. This warrants a more detailed explanation or at least a call for future research to address this discrepancy.

*Please note that the reconstructed changes in plant available moisture were affected by a mistake in the inputs used in the CO2 correction and this has now been corrected in the revised manuscript (see response to comment by Chris Brierley above). It is beyond the scope of this paper to determine why the various models show different trends in moisture availability through time. In the original text, we pointed out that climate models have difficulty both in simulating moisture transport into Europe and in capturing the land-surface feedbacks associated with changes in moisture availability. Differences in model configuration and parameterisations underpin both of these problems, but it would be difficult to determine this without sensitivity experiments. Our point here is that the differences between the modelled trends, and the relatively small simulated changes in the late Holocene, mean that the reconstructed changes in plant-available moisture cannot be simply a function of orbital forcing - which would tend to cause similar trends across the models. We have tried to make this clearer in the discussion as follows:*

We have shown that α was similar to today around 11 ka, but there was a rapid increase in moisture availability after ca 10.5 ka such that α values were noticeably higher than present between 10 to 6 ka, followed by a gradual and continuous decrease until the present time. Changes in the late Holocene are small even at centennial scale (Figure 5). The reconstructed trends in α are not captured in the simulations, which show different trends during the late Holocene. Thus, it is unlikely that the gradual increase in aridity during the late Holocene is a straight-forward response to orbital forcing. Changes in α in the EMBSeCBIO region are likely to be primarily driven by precipitation changes, which in turn are driven by changes in atmospheric circulation. Differences in the trend of moisture availability between the models imply that the nature of the changes in circulation varies between models and thus the simulations do not provide a strong basis for explaining the observed patterns of change in moisture availability. Earlier studies, focusing on the western Mediterranean (Liu et al., 2023), Europe (Mauri et al., 2014) and central Eurasia (Bartlein et al., 2017), have shown that models have difficulty in simulating the enhanced moisture transport into the Eurasian continent shown by palaeoenvironmental data during the mid-Holocene and during the late Holocene. Changes in precipitation can also affect land-surface feedbacks. Liu et al. (2023), for example, have argued that enhanced moisture transport into the Iberian peninsula during the mid-Holocene led to more vegetation cover and increased evapotranspiration and had a significant impact in reducing growing season temperatures. Differences in the reconstructed trends of summer temperature and plant-available moisture through the Holocene suggests that this land-surface feedback was not an important factor influencing summer temperatures in the EMBSeCBIO region. Nevertheless, differences in the strength of land-surface feedbacks between models could also contribute to the divergences seen in the simulations. It would be useful to investigate the role of changes in atmospheric circulation for precipitation patterns during the Holocene in the EMBSeCBIO region using transient simulations at higher resolution or regional climate models.

3. The authors also allude to the potential implications of their findings for understanding shifts in population density and cultural changes but do not expand upon this. It would be interesting to explore these implications further in the discussion, or in future work.

*We take this comment as an opportunity to expand on the potential implications of our reconstruction, particularly of our reconstruction of plant available moisture, for the origin of the agriculture in the region that lead to the expansion of population. In the discussion section, We have added the following paragraph in the discussion:*

The timing of the transition to agriculture in the eastern Mediterranean is still debated (Asouti & Fuller, 2012). It has been argued that climatic deterioration and population growth during the Younger Dryas triggered a shift to farming (Weiss & Bradley, 2001; Bar-Yosef et al., 2017). The presence of morphologically altered cereals by the end of the Pleistocene has been put forward as evidence for an early transition to agriculture (Bar-Yosef et al., 2017), but it has also been pointed out that the evidence for cereal domestication before ca 10.5ka is poorly dated and insufficiently documented (Nesbitt, 2002) and that crops did not replace foraging economies until well into the Holocene (Smith, 2001; Willcox, 2012; Zeder, 2011). The availability of water is a crucial factor in the viability of early agriculture (Richerson et al., 2001; Zeder, 2011). We have shown that moisture availability was higher than today during the first part of the Holocene (10-6 ka) but similar to today until ca 10. 5 ka. Wetter conditions during the early Holocene could have been a crucial factor in the transition to agriculture, and our findings support the idea that this transition did not happen until much later than the Younger Dryas or late glacial/Holocene transition. Further exploration of the role of climate in the transition to agriculture would require a more comprehensive assessment of the archaeobotanical evidence. The issue could also be addressed using modelling to explore how the reconstructed changes in regional moisture availability and seasonal temperatures would impact crop viability (see e.g. Contreras et al., 2019).

Contreras, D.A., Bondeau, A., Guiot, J., Kirman, A., Hiriart, E., Bernard, L., Suarez, R., Fader, M.: 2019. From paleoclimate variables to prehistoric agriculture: Using a process-based agro-ecosystem model to simulate the impacts of Holocene climate change on potential agricultural productivity in Provence, France, Quat. Internat., 501, 303-316.

4. While the authors have made efforts to correct for past CO2 changes, it would be beneficial to have a more detailed discussion about the potential implications of recent anthropogenic climate changes and their impacts on the trends identified in the study.

*Although we have not explicitly discussed the direct effect on anthropogenic changes in $CO_2$ on the reconstructions, it is essential to note that the $CO_2$ correction method already takes into account recent increases in atmospheric $CO_2$ levels. As described in the Prentice et al. (2022) model, the correction accounts for changes in water use efficiency in response to elevated $CO_2$ levels, leading to a more efficient water utilization by plants under current atmospheric conditions. To make this point clearer in the paper, we have modified the description of the correction procedure as follows:*

We applied a correction factor (Prentice et al., 2022) to the reconstructions of α to account for the impact of changes in atmospheric $CO_2$ levels on water-use efficiency, specifically the increased water use efficiency under high $CO_2$ levels characteristic of the recent past and the low $CO_2$ levels that would have reduced water use efficiency during the late glacial and thus could have influenced the reconstructions during the earliest part of the records.

5. The discussion section seems to be a heavy focus on comparing their findings with those of Mauri et al. (2015) and Herzschuh et al. (2022). While these comparisons are important, incorporating a broader range of studies would enrich the analysis and give readers a more comprehensive understanding of the state of research in this area.

*While there is a rich literature on palaeoenvironmental changes in the region, the only quantitative reconstructions currently available are those of Mauri et al. (2015) and Herzschuh et al. (2022). Our reconstructions are based on an improved data coverage and, we believe, a more robust methodology. We feel it is important to compare these reconstructions with previous reconstructions to highlight similarities and differences. Comparisons with the more qualitative palaeoclimate interpretations from individual sites would considerably expand the length and alter the focus of the paper.*

Minor comments.:
In Fig5-Fig7, the y-axis should include lowest/highest value, for example, L215 mentioned about 8°C lower, but y-axis only has 7.5°C.

*We have expanded the number of intervals on the scales, ensuring that they encompass both the minimum and maximum values of the smoothed mean, and using whole numbers exclusively. Please see response to a similar comment above.*

Line 223, "There was a gradual decrease in temperature after 5k", but in Fig.5 the decrease starts from around 4k.

*The pivotal moment in MTCO happens around 5ka, whereas in MTWA and GDD0, it takes place slightly later, closer to 4.5ka. Due to the uncertainty surrounding these dates, we have employed the prefix "circa" (ca) to indicate approximate timings. Nonetheless, we acknowledge that we haven't maintained this consistency throughout the entire text. To rectify this, we have replaced "5ka" with "ca. 4.5 ka."*

Line 796, Fig5, "The green line..." should be "The blue line"

*We have now amended this.*

---

## Author Comment (AC5)

**Response to RC3 Chenzhu Chen**

We thank to Chunzhu Chen for their comments. *Our specific responses are given below (in blue italic) and changes in wording are indicate in normal blue script.*

Correction of the impact of changing $CO_2$ levels on plant-available moisture reconstructions is a significant improvement that has been previously ignored in many climate reconstructions. However, I disagree with the authors' contention that atmospheric $CO_2$ levels affected moisture reconstructions only in the earliest part of the records. In fact, starting around 5 ka, there is a clear difference between the corrected and uncorrected results.

*In response to a comment by Chris Brierley, we have checked our calculations and found an error in the temperature inputs (please see response above). We have rectified this oversight by using the average temperature of the growing season for the α correction. We did not claim that atmospheric $CO_2$ levels only affected moisture reconstruction in the earliest part of the record and indeed we pointed out that the correction lead to an increase in reconstructed moisture levels during the late Holocene (line 235). Nevertheless, it is true that the largest difference between the corrected and uncorrected reconstructions occurs when CO2 levels are lowest. At 12ka, for example, the difference is nearly 0.2 whereas in the late Holocene the biggest difference is 0.08 at 4.8 ka. We have modified the text to clarify this (please see text given above).*

The 7.5 degree Celsius temperature anomaly during the early Holocene appears to be very large for MTCO reconstruction. The magnitude is much larger than that of existing reconstructions and simulation results. This should be discussed further.

*Although the reconstructed temperature anomaly is larger than in the previous reconstructions, the change in the TRACE-21K-II simulation is ca 5° and it is ca 3°in the LOVECLIM simulation. This suggest that our reconstructed change is not implausible. We attribute the discrepancies between our reconstruction and existing datasets primarily to methodological differences. We have modified the discussion to make this clearer as follows:*

We have shown that winter temperatures increased sharply between 10.3 and 9.3 ka, but then continued to increase at a more gradual rate through the Holocene. The increase of ca 7.5°C is of the same order of magnitude to the increase shown in the TRACE-21K-II simulation (ca. 5°C) and in the LOVECLIM simulation (3°C). This increasing trend is also seen in the Mauri et al. (2015) reconstructions of MTCO (Figure 9), although the change from the early Holocene to the present is much smaller (ca 0.5–1°C) in these reconstructions than in our reconstructions and Mauri et al. (2015) do not show marked cooling around 11 ka. Nevertheless, the consistency between the two reconstructions and between our reconstruction and the simulated changes in MTCO supports the idea that these trends are a response to orbital forcing during the Holocene.

It is understandable that the fossil pollen dataset (EMBSeCBIO; 20E-62E, 29N-49N) has a smaller spatial scale than the modern training dataset (20W-62E, 29N -75N), but it is difficult to understand why model outputs have a very different spatial range (EMBSeCBIO? 20W-55W, 29N-49N).

*This issue was raised by Chris Brierley. Please see response above.*

The manuscript includes simulation results from multiple models, but the explanation for the consistency, and especially the inconsistencies, between climate reconstruction and modeling needs to be strengthened. This study does not focus on simulations, but the results are presented in three separate figures. I recommend arranging the curves on a single large figure and, if necessary, including a figure to clearly compare the simulations and reconstructions.

*As we explain in responses to comments from the other reviewers, it is difficult to put the model results on a single figure or to combine these results with the reconstructions because of the differences in the magnitude of the trends. Our focus in the paper is to use the similarity in the trends, rather than in the magnitudes, to address whether that the forcings incorporated in the different simulations (orbital, greenhouse gas, ice sheets and meltwater) provide an explanation for the observed trends in the reconstructions. For the late Holocene, we also point out that the orbital forcing is insufficient to explain the trend in the reconstruction of plant available water since the models disagree on the direction of the trend. We have amplified our discussion of the similarities and differences between the simulations and reconstructions in response to comments from the other reviewers (please see changes to text above).*

Figure 9 lacks a figure legend.

We have added a legend inside the plot

---

## Author Response (AR2)

**Response to Referee #3: Chunzhu Chen**

We thank to Chunzhu Chen for their comments. We appreciate the acknowledgment of our work's contribution to understanding Holocene paleoclimate in the Eastern Mediterranean region. *Our specific responses are given below (in blue italic) and changes in wording are indicate in normal blue script.*

The manuscript presents a climate reconstruction spanning the Holocene period based on a collection of 71 pollen records from the Eastern Mediterranean region. It incorporates climate simulation results from multiple models to find out the forcings underlying climate variations. It makes an excellent contribution to our understanding of paleoclimate in the Eastern Mediterranean region. The region is fascinating because of its long human history, which has had a significant impact on vegetation and, as a result, the Holocene pollen records. While Cultural indicators have been removed from the fossil pollen dataset, pollen taxa that indicate both natural and anthropogenic impact pose a significant challenge in distinguishing the cause of vegetation changes. For instance, when humans clear forests for agriculture, we typically see a substantial decline in tree pollen and an increase in landscape openness, which can also be caused by cooling and/or drying. This has been observed since the middle Holocene, and may have occurred earlier in places such as Israel. The fossil pollen data used in this study more or less has human imprints, and many of these records open at ca. 5 ka. This is an important point to consider before using pollen records from the Eastern Mediterranean region to reconstruct Holocene climate variations. Aside from this, I believe the manuscript makes an important contribution, and minor revisions are required.

*As the reviewer acknowledges, we removed cultivation-related taxa from the training and the fossil data sets in order to minimise the impact of an anthropogenic signal in the records. We agree that human activities can also cause an increase in landscape openness, mimicking the impact of cooling and/or drying on the natural vegetation. However, we have examined the relationship between changes in forest cover, human population and climate changes in the Eastern Mediterranean region in a separate paper on the vegetation history of the region that is currently under review in Journal of Biogeography. This work shows (see figure below) that changes in forest cover from the mid-Holocene onwards are not closely related to the changes in population density, derived as summed probability distribution (SPD) of calibrated archaeological radiocarbon dates. They are, however, closely related to changes in moisture availability as indicated by oxygen isotope records from speleothems in the region. These findings suggest that the changes in forest cover during the late Holocene can be primarily attributed to climatic factors rather than human activities. This additional evidence provides confidence that our climate reconstructions are not substantially influenced by landscapes changes due to anthropogenic activities.*

*Archaeological evidence suggests that there is considerable variation in the timing of population changes in this region. It is widely accepted that the most significant population boom occurred during the Archaic-Classic and Roman periods, approximately 2,800 years ago (Palmisano et al., 2017; Stoddart et al., 2019) i.e. much later than the transition evident in our reconstructions at ca. 5ka. An independent assessment (Palmisano et al. 2021) of population density trends within subregions of the Eastern Mediterranean region shows that there was an overall increase in population across the whole region during the Late Pleistocene-Early Holocene ( 14 to 8.3 ky), but there were very different trends in different sub-regions during the Middle and Late Holocene (8.3 to 2.5 ky). These findings do not support a generalized population increase around ca. 5 ka, corresponding to the mid-Holocene transition evident in our climate reconstructions.*

*Finally, as we point out in the paper, our reconstructions of climate variables exhibit stronger congruence with climate model simulations driven by changes in external forcing during the late Holocene. Since these model simulations do not include changes in land use, this provides further*

*support for our interpretation that our climate reconstructions are robust and consistent with natural climatic forcings.*

[Figure]

*In response to this comment, we have revised the methods section to explain the purpose of removing cultivars more clearly as follows:*

Line 84: "The SMPDS pollen records have been taxonomically standardized, filtered to remove obligate aquatics, insectivorous species, introduced species, and taxa that only occur in cultivation. The removal of cultivars is designed to minimise the influence of anthropogenic signals on the reconstructions. We then grouped taxa with only sporadic occurrences into higher taxonomic levels

(genus, sub-family or family). Consequently, the data set provides relative abundance data for 247 pollen taxa (Supplementary Table 1).

Records from seas, such as the Black Sea, are excluded from the fossil pollen data. The seas are huge traps of pollen grains produced on land. Comparing to terrestrial pollen records, these records represent vegetation changes on a larger spatial scale and thus fit well with the manuscript's topic."

*We deliberately excluded marine records because they sample an extremely large area. In the case of the Black Sea, this could potentially be much of Europe and Russia, and thus inclusion of these records could potentially bias the reconstructions of Eastern Mediterranean climate. In fact, when we compare regional reconstructions made including the Black Sea records with those presented in the paper, the inclusion of marine records does not substantially alter the shape or magnitude of the reconstructed trends in climate. This indicates that, although from a pollen source theory point of view it makes sense to exclude marine records in making site-based climate reconstructions, the impact of this exclusion on our regional reconstructions appears to be limited.*

*The inclusion of records from the Black Sea was one of the methodological differences between our reconstructions and the Herzschuh et al reconstructions that we pointed out in our Discussion. Since inclusion of sites from the Black Sea does not affect our reconstructions substantially, we have modified this text as follows:*

Line 351: Specifically, Herzschuh et al. (2022) used (1) a unique calibration data set for each fossil site based on modern samples within a 2000 km radius of that site, rather than relying on a single training data set; (2) a limited set of 70 dominant taxa rather than the whole pollen assemblage; and (3) included marine records from e.g. the Black Sea, which were excluded in the other reconstructions because they sample an extremely large area and thus are unrepresentative of the local climate. However, inclusion of records from the Black Sea in our reconstructions does not have a substantial impact on either the magnitude or the trends in climate. Thus, it seems likely that the differences between these two reconstructions reflects the use of a unique calibration data set for each fossil site and the limited set of taxa included.